# Sex Hormones and Hormone Therapy during COVID-19 Pandemic: Implications for Patients with Cancer

**DOI:** 10.3390/cancers12082325

**Published:** 2020-08-18

**Authors:** Carlo Cattrini, Melissa Bersanelli, Maria Maddalena Latocca, Benedetta Conte, Giacomo Vallome, Francesco Boccardo

**Affiliations:** 1Department of Internal Medicine and Medical Specialties (DIMI), School of Medicine, University of Genoa, 16132 Genoa, Italy; carlo.cattrini@gmail.com (C.C.); mariamaddalena.latocca@gmail.com (M.M.L.); bntconte@gmail.com (B.C.); giacomo.vallome@gmail.com (G.V.); 2Prostate Cancer Clinical Research Unit, Spanish National Cancer Research Centre (CNIO), 28029 Madrid, Spain; 3Medical Oncology Unit, University Hospital of Parma, 43126 Parma, Italy; bersamel@libero.it; 4Academic Unit of Medical Oncology, IRCCS Ospedale Policlinico San Martino, 16132 Genoa, Italy

**Keywords:** COVID-19, SARS-CoV-2, estrogens, androgens, progesterone, testosterone, TMPRSS2, ACE2, camostat, androgen-deprivation therapy, tamoxifen

## Abstract

The novel coronavirus disease 2019 (COVID-19) shows a wide spectrum of clinical presentations, severity, and fatality rates. The reason older patients and males show increased risk of severe disease and death remains uncertain. Sex hormones, such as estradiol, progesterone, and testosterone, might be implicated in the age-dependent and sex-specific severity of COVID-19. High testosterone levels could upregulate transmembrane serine protease 2 (TMPRSS2), facilitating the entry of severe acute respiratory syndrome coronavirus 2 (SARS-CoV-2) into host cells via angiotensin-converting enzyme 2 (ACE2). Data from patients with prostate cancer treated with androgen-deprivation therapy seem to confirm this hypothesis. Clinical studies on TMPRSS2 inhibitors, such as camostat, nafamostat, and bromhexine, are ongoing. Antiandrogens, such as bicalutamide and enzalutamide, are also under investigation. Conversely, other studies suggest that the immune modulating properties of androgens could protect from the unfavorable cytokine storm, and that low testosterone levels might be associated with a worse prognosis in patients with COVID-19. Some evidence also supports the notion that estrogens and progesterone might exert a protective effect on females, through direct antiviral activity or immune-mediated mechanisms, thus explaining the higher COVID-19 severity in post-menopausal women. In this perspective, we discuss the available evidence on sex hormones and hormone therapy in patients infected with SARS-CoV-2, and we highlight the possible implications for cancer patients, who can receive hormonal therapies during their treatment plans.

## 1. Sex-Specific and Age-Dependent Severity of Coronavirus Disease 2019 (COVID-19)

Increasing data support the notion that older age and male gender are factors associated with a significantly increased risk of severe events and death from coronavirus disease 2019 (COVID-19) [1,2,3]. Of the 1591 patients admitted to intensive care units (ICU) in the Lombardy region, Italy, 82% were male [4]. Another Canadian study analyzed over 200,000 residents tested for severe acute respiratory syndrome coronavirus 2 (SARS-CoV-2); although only 36% of people tested were males, compared with women, they showed higher rates of laboratory-confirmed infection (13.5% vs. 9.8%), hospitalization (15.6% vs. 10.4%), ICU admission (4.1% vs. 1.7%), and death (8.7% vs. 7.6%) [5]. In the U.K. COVID Symptom Study, data from more than 2.5 million users of the COVID Symptom Tracker App were analyzed [6]. This study further confirmed that older age and comorbidities were associated with increased odds of requiring hospital admission, and men with COVID-19 were more likely than women to need respiratory support (odds ratio: 2.14 (95% confidence interval (CI): 1.72–2.66)). A sex bias in COVID-19 mortality is currently reported in 37 of the 38 countries that have provided sex-disaggregated data [7].

The variable expression of angiotensin-converting enzyme 2 (ACE2), which mediates SARS-CoV-2 binding and entry into cells, has been proposed to explain the age and gender gap [8,9,10]. However, other studies have not found a substantial difference in ACE2 expression between males and females, nor between younger and older women [11,12]. An Italian study suggested that ACE2 variants and genetic background may contribute to explain the inter-individual clinical variability associated with COVID-19 [13]. Of note, it is not universally accepted that high ACE2 levels are deleterious. A higher density of ACE2 expression might be beneficial when SARS-CoV-2 competes with angiotensin II for binding sites [14], and mounting evidence suggests that SARS-CoV-2 downregulates ACE2 expression [15]. Low levels of ACE2 and high levels of angiotensin II might lead to increased pulmonary vessel permeability, which in turn could result in the inflammatory damage of lungs that characterize critically ill patients [16]. This theory is consistent with the observation that basal ACE2 expression is increased in children, who show better outcomes, and reduced in patients with type II diabetes who are at significant risk of severe COVID-19. In addition, ACE2 is downregulated by inflammatory cytokines and upregulated by estrogen and androgen, which both decrease with age—a possible explanation of the age-dependent severity [17].

Several other reasons have also been suggested to explain the gender disparity in COVID-19, such as the different habit of smoking and drinking, sociological, psychological factors, and the different profile of comorbidities among sexes [18]. Importantly, some of these factors have been related to a differential ACE2 expression [17,19].

The complex mechanisms that underlie the variability of immune responses according to age and sex might also explain the age-dependent and sex-specific severity of COVID-19 [7,20]. The immune-senescence modifies the extent and strength of pathogen clearance during infections, and this mechanism might contribute to explain the age-dependent severity of COVID-19 [21]. In addition, distinct immune responses are demonstrated between the sexes, and can result in differential incidence and susceptibility of males and females to autoimmune diseases, tumors, infections, and response to vaccines [21]. A recent study suggests that women with severe COVID-19 have a higher concentration of serum SARS-CoV-2 IgG antibody compared with men, and the generation of IgG antibodies in the early phases of infection seems to be stronger in females than in males [22]. It is known that estrogens, androgens, and the sex chromosome complement significantly affect immune responses in both sexes [23,24]. In addition, mounting evidence suggests that both sex hormones and hormone therapy might be useful in COVID-19 through mechanisms of direct antiviral activity or immune modulation [25] (Figure 1).

## 2. Lessons from 2002–2003 SARS-CoV

Although few experimental data are currently available on SARS-CoV-2, experiences with analogous coronaviruses could be helpful. Similar to SARS-CoV-2, the 2002–2003 severe acute respiratory syndrome coronavirus (SARS-CoV) and the middle east respiratory syndrome coronavirus (MERS-CoV) showed age-dependent and sex-specific severity [26,27,28,29,30]. In addition, SARS-CoV and SARS-CoV-2 are both RNA β-coronaviruses that share similar clinical presentations, demographic characteristics, and laboratory and radiological findings [31,32]. SARS-CoV-2 is almost 80% similar to SARS-CoV at a nucleotide level, and the receptor binding domains of both viruses bind to the human ACE2 receptor [33,34]. However, different from SARS-CoV, more frequent cases of asymptomatic or mild-moderate SARS-CoV-2 are described [31].

Similar to that observed in the patients infected with SARS-CoV-2, in vivo experiments with SARS-CoV showed that mice underwent age-dependent and sex-specific susceptibility to this infection [35]. Increased inflammatory monocyte-macrophages, neutrophils, and proinflammatory cytokines/chemokines were found in the lungs of SARS-CoV infected male compared to female mice. In females, gonadectomy was associated with progressive weight loss and death after SARS-CoV infection, and with an increased accumulation of monocyte-macrophages in the lungs. In addition, increased susceptibility to SARS-CoV was found in mice treated with an estrogen receptor (ER) antagonist compared with those treated with tamoxifen or the control. More importantly, weight loss was significantly reduced in mice treated with the ER modulator tamoxifen compared with the control or ER antagonist. In males, gonadectomy or treatment with the antiandrogen flutamide did not affect morbidity and mortality, but a significant decrease in the serum testosterone levels was found during infection. The authors concluded that ER signaling was important to protect female mice from SARS-CoV infection, whereas androgens did not influence disease outcome in males. The protective role of estrogens has also been demonstrated in experiments, including mice infected with influenza virus [36]. In this study, infected male mice showed reduced survival after the removal of the gonads.

An intriguing mechanistic explanation of a potential protective effect induced by sex hormones against the worst consequences of SARS-CoV-2 infection comes from data on SARS-CoV [37]. The authors demonstrated that the viral spike protein induces macrophages to produce proinflammatory cytokines (IL-6 and TNF-α) via activation of the nuclear factor κB (NF-κB). This protein complex is also activated by angiotensin II to induce lung vasoconstriction and inflammation [38], whereas it is inhibited by androgens and estrogens. Therefore, sex hormones could prevent cytokine release by blocking NF-κB, which is activated by angiotensin II in the absence of its neutralizing enzyme, ACE2, which in turn is downregulated by SARS-CoV-2 [39].

Other experiments and network-based analyses of drug repurposing showed that tamoxifen and toremifene exert an antiviral activity against MERS-CoV and SARS-CoV, and could be helpful in SARS-CoV-2 [40,41]. Similar in vitro antiviral effects on SARS-CoV and MERS-CoV have also been described with protease inhibitors targeting the transmembrane serine protease 2 (TMPRSS2), an androgen-regulated gene commonly known in prostate cancer biology [42].

## 3. Androgens: A Double-Edged Sword?

Several studies suggest that both high and low testosterone levels could favor severe COVID-19 [43,44]. A recent analysis about the epidemiology of SARS-CoV-2 in the Italian Veneto region supports the hypothesis that androgen-deprivation therapy (ADT) might protect men from infection [45]. In this study, prostate cancer patients who were not receiving ADT showed an increased risk of SARS-CoV-2 infection compared with those who were receiving ADT (OR 4.05; 95% CI, 1.55–10.59). Of the 5273 patients with prostate cancer receiving ADT, only four were found to be positive for SARS-CoV-2, compared with the 114 positives of the 37,161 patients with prostate cancer who were not receiving ADT. Nevertheless, the difference between the two groups might be due to different exposures to SARS-CoV-2. In addition, this study included a cohort of patients identified on a regional cancer registry, who were receiving ADT for different settings (adjuvant, biochemical relapse, and metastatic disease); given that the disease stage was not reported, a significant selection bias might have impacted on these results. Indeed, the postulated protective role of ADT seems to be resized in patients with advanced disease, according to the results of another Italian report that only included metastatic prostate cancer patients [46]. Moreover, the recent OnCovid study on 890 cancer patients infected with SARS-CoV-2 reported that patients with genitourinary tumors had a significantly worse median survival (22.0 months (95% CI, 5.3–36.6)) compared with those with other neoplasms, further calling into question the putative protective role of ADT [47]. On the other hand, two Spanish studies reported that a significant percentage of patients admitted to ICU suffered from androgenetic alopecia, supporting the androgen-driven COVID-19 theory [48,49,50]. A mechanistic explanation for this theory is related to the viral entry mediated by TMPRSS2 (Figure 1A). The TMPRSS2-dependent pathway has been reported to be the predominant route in lung alveolar and gastrointestinal cells, whereas the route of entry in other cells, such as vascular and endothelial cells, remains unknown [51]. This gene is also the most frequently altered in primary prostate cancers [52], and TMPRSS2 expression is strongly upregulated in response to androgens in prostate cancer cells and in human lung epithelial cell lines [53,54]. A transcriptomic analysis of the available datasets has not revealed substantial differences in TMPRSS2 expression between males and females or according to age [11]; however, the authors of this analysis speculated that specific genetic variants could induce higher levels of TMPRSS2 and could explain different fatality rates among countries. Preclinical data suggest that camostat mesylate, which inhibits the protease activity of TMPRSS2, is able to block the entry of SARS-CoV-2 in lung epithelial cells [51,55]. Clinical studies are ongoing to determine whether camostat or other TMPRSS2 inhibitors, such as nafamostat or bromhexine, could improve the outcomes of men with COVID-19 (Table 1) [52,56]. Studies are also investigating the use of ADT or antiandrogens, such as bicalutamide or enzalutamide, to prevent complications of COVID-19. Of note, as a relationship between testosterone and thrombosis has been previously described [57], high testosterone levels could also contribute to the development of microthrombi and venous thromboembolism, which are characteristics of patients with severe COVID-19 [58]. Other preclinical data also support the assumption that the five alpha reductase inhibitors, such as dutasteride, could be helpful in COVID-19, by reducing the ACE2 levels and the internalization of the spike receptor binding domain [59].

Although the androgen-driven theory is intriguing, it remains unexplained why younger men with COVID-19, who have higher testosterone levels compared with older men, show reduced severity and case-fatality rates [60]. Similarly, it would be unexpected that older men who have lower testosterone levels show increased severity and fatality rates to COVID-19.

Beyond the possible implications as a result of the regulation of TMPRSS2, male hormones significantly modulate both innate and adaptive immunity [21,61] (Figure 1B). Androgens generally suppress the inflammatory responses by decreasing the activity of the peripheral blood mononuclear cells, as well as the release of inflammatory factors and cytokines, such as interleukin-1β, interleukin-2, tumour necrosis factor-α, inducible nitric oxide synthase, and nitric oxide [21,62,63]. They can also promote the synthesis of suppressive cytokines such as interleukin-10 and the transforming growth factor-β through the androgen receptor signaling [63,64,65]. In addition, androgens are involved in the thymic involution after puberty [61]—thymic involution diminishes the export of T cells towards the periphery, and reduces the function of peripheral T cells and the repertoire of T cell receptors. It is not surprising that high testosterone levels have been associated with a poor virus-neutralizing activity after influenza vaccination [66]. These immunosuppressive effects of androgens might promote SARS-CoV-2 infection, but might also inhibit the cytokine storm that characterizes the most severe cases of COVID-19. An Italian study on 31 male patients admitted to ICU found that lower testosterone plasma levels predicted worse outcomes, supporting a favorable role of androgens during SARS-CoV-2 infection [67]. Similarly, a retrospective German study reported that critically ill male patients with COVID-19 suffered from severe testosterone and dihydrotestosterone deficiencies [68]. Another Chinese study showed that the ratio of testosterone to luteinizing hormone was significantly reduced in patients with COVID-19 [69]. These data support the notion that reduced testosterone levels in pre-existing or potentially virus-induced hypogonadism might be associated with an adverse prognosis in COVID-19 [70]. In summary, it remains currently unclear whether androgens might be favorable or unfavorable during SARS-CoV-2 infection. Further investigations are warranted to assess the relationship between hypogonadism and COVID-19, as well as the putative protective role of ADT or of other antiandrogen treatments against severe disease.

## 4. Estrogens and Progesterone: A Protective Role?

As previously reported, estrogens were found to be crucial to protect female mice from SARS-CoV infection [35]. The observation that women with COVID-19 show better outcomes compared to men and that post-menopausal women are those at higher risk of severe COVID-19 is consistent with the possibility that estrogens could protect females from severe COVID-19. In vitro data suggest that estrogens can downregulate the expression of ACE2 mRNA, but not TMPRSS2, in bronchial epithelial cells [71]. Beyond this mechanism, the potential favorable role of estrogens might also be related to their immune-modulating properties (Figure 1B). Progesterone and 17β-estradiol (E2) have distinct roles in modulating innate and adaptive immunity [21]. E2 shows bipotential effects based on concentration, and it is able to facilitate the progression of the inflammatory processes toward a deactivated phenotype, restoring homeostatic conditions [72]. Low concentrations of E2 enhance the production of pro-inflammatory cytokines, and promote TH1-type responses and cell-mediated immunity, whereas high concentrations of E2 reduce the secretion of pro-inflammatory cytokines, and increase TH2-type responses and humoral immunity [73]. E2 was critical to promote the recruitment of neutrophils in a mouse model, favoring the response of CD8 T cells against influenza A virus [74].

Progesterone generally promotes anti-inflammatory responses and can favor the skewing of CD4+ T cell from TH1-type towards TH2-type responses [75]. It has been proposed that an immune system prone to a TH2-type response, such as in patients with asthma, might protect against severe COVID-19 [76]. In addition, progesterone has a partial glucocorticoid and mineralocorticoid activity. Given that steroidal treatment is beneficial and is used in clinical practice for some cases of severe COVID-19 [77], the activation of the steroid receptor by progesterone might also contribute to mitigate excessive immune responses [78]. Finally, recent data suggest that progesterone could exert a direct antiviral activity on SARS-CoV-2 through the modulation of the Sigma receptors [79].

An analysis adjusted for age and severity in 1902 women with COVID-19 in Wuhan reported that post-menopausal patients showed longer hospitalization times compared with non-menopausal women (HR: 1.91 (95% CI, 1.06–3.46), *p* = 0.033) [80]. High levels of E2 and of anti-müllerian hormone were associated with reduced disease severity in logistic regression (HR: 0.30 (95% CI, 0.09–1.00), *p* = 0.05; HR: 0.15 (0.03–0.82), *p* = 0.03) and negatively correlated with IL-2R, IL-6, IL-8, and TNFα levels in the luteal phase (Pearson = −0.592, −0.558, −0.545, and −0.623; *p* = 0.033, 0.048, 0.054, and 0.023). Higher levels of IL6 and IL8, in turn, were found in patients with severe disease (*p* = 0.040, 0.033).

The team of the COVID Symptom Study recently analyzed the data collected with the COVID Symptom Tracker Application in the U.K. (152,637 women for menopause status, 295,689 for oral contraceptive use, and 151,193 for hormone-replacement therapy use) [81]. After adjustment for age, smoking, and body mass index, post-menopausal women aged 40–60 years had a higher rate of symptom-based predicted COVID-19 (*p* = 0.003) and consistent trends for tested COVID-19, and requirement for hospitalization and respiratory support compared with pre-menopausal women. In addition, women aged 18–45 years taking oral contraceptives had a significantly lower predicted COVID-19 (*p* < 0.001), with a reduction in hospital attendance (*p* = 0.023). Post-menopausal women using hormonal therapies did not show consistent associations, except for increased rates of predicted COVID-19 (*p* < 0.001) for those who were receiving hormone-replacement therapy. However, this last result was potentially biased by a lack of information on the route of administration, comorbidities, and the type and duration of hormonal treatment. Therefore, the authors concluded that their findings supported the notion that estrogens could have a protective effect on COVID-19 [81].

The ability of progesterone and estradiol to reduce severity in men with COVID-19 is being tested in clinical trials and the results are pending (Table 1). As previously mentioned, in vitro and in vivo data from MERS-CoV and SARS-CoV also suggest that selective estrogen receptor modulators, such tamoxifen and toremifene, might be helpful against COVID-19 [35,40,41], further reinforcing the need of more investigations in patients treated with these agents.

## 5. Effect of Hormone Therapy in Patients with Cancer and COVID-19

Except for the analysis of the potential protective role of ADT in patients with prostate cancer previously described [45], scanty data are currently available in patients affected by COVID-19 receiving endocrine therapy for cancer. A New York study on 218 patients with cancer and COVID-19 suggested that genitourinary and breast malignancies were associated with a relatively lower mortality during SARS-CoV-2 infection compared with other tumors [82]. However, no data on the use of hormone therapy were available. The COVID-19 and Cancer Consortium registry database provided the outcome data on 928 patients with confirmed SARS-CoV-2 infection and active or previous malignancy from the USA, Canada, and Spain [83]. The most prevalent tumors were breast (21%) and prostate (16%). The authors demonstrated that age, male sex, smoking status, number of comorbidities, performance status, presence of active cancer, and receipt of azithromycin plus hydroxychloroquine were all independent factors associated with increased 30-day mortality. The cancer type (solid vs. hematological vs. multiple) or type of anticancer therapy were not associated with mortality. However, only 85 patients (9%) were receiving endocrine therapy, and the receipt of hormonal treatment was not investigated as a potential prognostic variable. The U.K. Coronavirus Cancer Monitoring Project prospectively described the clinical and demographic characteristics of 800 patients with a diagnosis of cancer and symptomatic COVID-19 [84]. In the multiplicity unadjusted, univariate regression, breast and female genital cancers were associated with a lower mortality from COVID-19 (odds ratios (OR): 0.48 (0.28–0.84), *p* = 0.009; 0.31, (0.11–0.81), *p* = 0.010), whereas male genital tumors were associated with increased mortality (OR: 1.99 (1.14–3.48) *p* = 0.015). However, after Bonferroni *p* value adjustment, only advanced age, male sex, and the presence of other comorbidities (hypertension and cardiovascular disease) were identified as significant risk factors for death. The receipt of chemotherapy in the past four weeks did not affect mortality from COVID-19 in the multivariate model. For all tumors, only 64 patients (8%) were receiving hormone-therapy within 4 weeks from COVID-19 diagnosis, of these 21 (33%) died. Hormone therapy was not identified as a prognostic predictor for death in the multivariate model, but this result was inconclusive due to a low statistical power (odds ratio: 0.90 (0.49–1.68), *p* = 0.74). The multicenter OnCovid study assessed the outcome of 890 cancer patients with confirmed COVID-19 [47]. Male gender, older age, and co-morbidities identified a subgroup of patients with significantly worse mortality rates from COVID-19. Use of chemotherapy, targeted therapy, or immunotherapy did not worsen mortality. The receipt of hormone-therapy was not associated with mortality, but inadequate power also affected this analysis (HR: 1.20, (95% CI, 0.71–2.04), *p* = 0.48). Another retrospective, multicenter study analyzed the outcome of 205 cancer patients with SARS-CoV-2 infection in Hubei region, China [85]. The authors reported that recent use of chemotherapy and male sex were risk factors for death during hospital admission. However, of the 182 assessable patients, only 10 were receiving endocrine therapy or traditional Chinese medicine, and none of these patients died.

## 6. How Could These Data Influence the Daily Clinical Practice?

Hormone therapy is a common treatment in patients with prostate and breast cancers. Similarly, patients with ovarian or uterine cancer often undergo radical surgery that prematurely compromises the ovarian function and the maintenance of physiological hormonal levels. Testosterone deficiencies are also commonly observed in patients with testicular cancer who receive bilateral orchiectomy. The unanswered question is whether these therapeutic interventions might affect the outcome of cancer patients infected with SARS-CoV-2. As previously mentioned, epidemiological data support that ADT might exert a protective role in patients with prostate cancer [45]. Preclinical data showed that inhibitors of TMPRSS2 and of 5 alpha reductase might be active against SARS-CoV-2 [51,55,59]. In vivo data on SARS-CoV also suggested that estrogen antagonists might be deleterious during infection, whereas the selective estrogen receptor modulators might be favorable [35]. In the COVID Symptom Study, post-menopausal women had a higher rate of symptom-based predicted COVID-19 compared with pre-menopausal women, and use of oral contraceptives seemed to protect from SARS-CoV-2. However, this positive association was not detected in post-menopausal patients who were taking hormone-replacement therapy. Although the low statistical power represents a significant limit, the clinical studies conducted so far have not identified that endocrine therapy might affect the prognosis of cancer patients with COVID-19. Currently, there is no sufficient evidence to support that male or female hormone deprivation or administration, or specific hormonal agents, could positively or adversely affect the outcome of patients with cancer infected with SARS-CoV-2. Therefore, no change in the hormonal management of these patients is currently recommended [86]. However, a strong rationale supports the potential role of sex hormones and hormone therapy in modulating the clinical course during infection from SARS-CoV-2. Given their expected long-term survival, breast cancer patients treated with adjuvant hormone-therapy could be good candidates to understand the effects of either estrogen deprivation or estrogen modulation. Studies on the prevalence and severity of SARS-CoV-2 in young patients who underwent radical ovarian surgery could also be helpful. Finally, further studies are warranted to assess whether high or low testosterone levels could affect the outcomes of men with COVID-19, and whether ADT or other treatments might favorably affect the outcome of patients with prostate cancer.

## 7. Conclusions

Emerging data suggest that sex hormones are key actors in the age-dependent and sex-specific severity of COVID-19, through several mechanisms, such as regulation of the immune responses, modulation of SARS-CoV-2 cell entry, and affecting hemostatic function. The literature data are consistent with a potential protective role of female hormones, whereas conflicting results are reported regarding androgens. Chemotherapy, targeted therapy, and immunotherapy do not seem to affect the fatality rates of cancer patients after SARS-CoV-2 infection. However, further investigations are needed to assess the effects of hormone-therapy and hormone-deprivation in male and female patients with cancer, given their potential implications in modulating the severity and fatality of COVID-19.

## Figures and Tables

**Figure 1 cancers-12-02325-f001:**
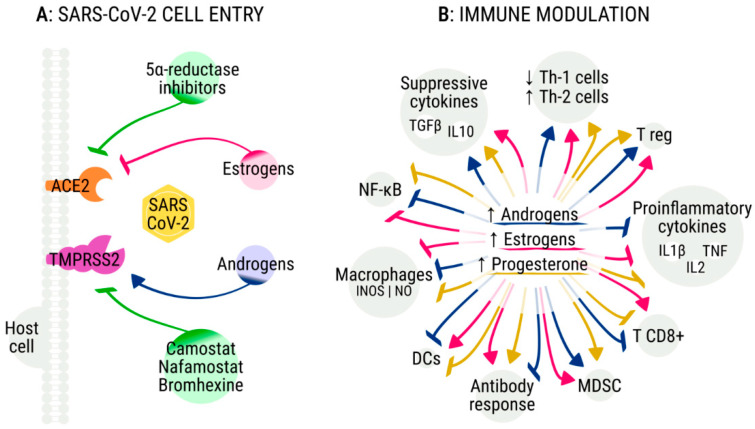
The role of sex hormones and hormone therapies in modulating severe acute respiratory syndrome coronavirus 2 (SARS-CoV-2) entry in host cells and immune response. (**A**) A proposed model suggests that androgens can upregulate the activity of transmembrane serine protease 2 (TMPRSS2), which is necessary for the SARS-CoV-2 spike protein priming; several TMPRSS2 inhibitors are under investigation in clinical trials. In vitro data also suggest that estrogens might downregulate the angiotensin-converting enzyme 2 (ACE2) expression, which is used by SARS-CoV-2 for host cell entry. (**B**) Androgens (blue arrows), estrogens (pink arrows), and progesterone (yellow arrows) can activate and inhibit several components of the immune response against SARS-CoV-2, affecting the clinical course and disease severity of patients with COVID-19. TGFβ—transforming growth factor-beta; IL—interleukin; Th-1—T helper 1; Th-2—T helper 2; T reg—T regulatory cells; TNF—tumor necrosis factor; MDSC—myeloid-derived suppressor cells; DCs—dendritic cells; INOS—inducible nitric oxide synthase; NO—nitric oxide; NF-κB—nuclear factor kappa-light-chain-enhancer of activated B cells.

**Table 1 cancers-12-02325-t001:** On-going studies of hormone therapies and TMPRSS2 inhibitors in coronavirus disease 2019 (COVID-19) patients.

Study ID	Treatment	Population	Estimated Enrollment	Phase	Randomized	Primary Endpoint	Estimated Completion Date
NCT04359329	Estradiol patch	Male ≥ 18 years Females ≥ 55 years	110	2	Yes Open label	Hospitalization ICU transfer Intubation Death	11/2020
NCT04389580	Tamoxifen + Isotretinoin	Patients with severe respiratory failure requiring transfer to ICU	160	2	Yes Open label	Lung injury score	09/2020
NCT04365127	Progesterone	Male patients with documented pneumonia	40	1	Yes Open label	Change in clinical status at day 15	04/2021
NCT04374279	Bicalutamide	Patients requiring hospitalization but with minimal respiratory symptoms	60	2	Yes Open label	Clinical improvement at day 7	06/2021
NCT04456049	Enzalutamide	High risk outpatient adult males with confirmed COVID-19	90	2	Yes Open label	Decrease the nasopharyngeal swap SARS-CoV-2 viral load	12/2021
NCT04475601	Enzalutamide	Mild to severe symptoms of COVID-19	500	2	Yes Open label	Time to worsening of disease	08/2022
NCT04397718	Degarelix	Male veterans hospitalized for COVID-19	198	2	Yes Placebo-controlled	Mortality (day 15)	10/2020
NCT04338906	Camostat	Hospitalized patients	334	4	Yes Placebo-controlled	Hospital discharge (day 14)	12/2021
NCT04355052	Camostat	Patients with mild/moderate disease	250	3	Yes Open label	Clinical state as per NEWS scoring (day 7) PCR positivity (day 7)	12/2020
NCT04353284	Camostat	Outward patients	114	2	Yes Open label	Viral load (day 2)	05/2021
NCT04321096	Camostat	Hospitalized patients	580	1/2	Yes Placebo-controlled	Hospital discharge or clinical improvement	05/2021
NCT04374019	Camostat	High-risk patients	240	2	Yes Multi-arm	Clinical deterioration	05/2021
NCT04352400	Nafamostat	Hospitalized patients	256	2/3	Yes Placebo-controlled	Time to hospital discharge or clinical improvement	12/2021
NCT04418128	Nafamostat	Hospitalized patients with documented pneumonia	84	2/3	Yes Open Label	Hospital discharge or clinical improvement	04/2021

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
