# Peer review of "Sex Hormones and Hormone Therapy during COVID-19 Pandemic: Implications for Patients with Cancer"

_cancers, 2020, doi:10.3390/cancers12082325_

Round 1
Reviewer 1 Report
I think that in the resubmitted version the paper is acceptable in its contents and scientific quality. Authors make changes and integration as requested inj my previous review and I think now paper is suitable for publication even if I think some other little modifications have to be made particularly in english language and in the Conclusion that in my opinion have to be integrated and have to better summarize the results
Reviewer 2 Report
The authors have sufficiently addressed reviewer's comments. The manuscript has been substantially improved after revision.
The paper should be considered for publication in Cancers.
Reviewer 3 Report
The revision address my comments adequately.
This manuscript is a resubmission of an earlier submission. The following is a list of the peer review reports and author responses from that submission.
Round 1
Reviewer 1 Report
In my opinion Authors perform a sufficently complete revision of literature on ormones influence in clinical presentation of Sars-Cov 2 desease. I think therefore that research is limited to described results of study conducted in the past both on Sars-Cov 2 desease and Sars Cov desease (with some reference even to Mers Cov desease). I think that the literature review that Authors perform in the paper, to be useful for publication, would need of own experimental data, to make paper suitable for publication. in fact the paper without any experimental data seems to be lacking of innovative observations and limited only to expose results of previous study.
Reviewer 2 Report
In this paper, the authors report the effects of sex hormones and treatments on COVID-19 severity. Although the effects of these factors on the pathogenesis of the disease have not been conclusively determined at present, various case reports have been discussed from various perspectives and are considered useful for future treatment methods.
While the authors discuss the importance of TMPRSS2 in SARS-CoV-2 infection and focus on the androgenic signals involved in its induction, there are two major routes of entry of SARS-CoV-2. One is a TMPRSS2-dependent pathway from the plasma membrane and the other is a cathepsin-dependent pathway through endocytosis. A TMPRSS2-dependent pathway has been reported to be the predominant one in lung alveolar and gastrointestinal cells, but the route of entry in other cells, such as vascular endothelial cells, remains unknown. Therefore, the authors should mention cell-specific effects in discussing the importance of TMPRSS2.
Minor points:
line 65, SARS-CoV-2, The hyphen is missing.
line 135, COVID-19, "19" is missing.
Reviewer 3 Report
Sex hormones and hormone therapy during COVID-19 pandemic: implications for patients with cancer by Cattrini et al.
The report is well written and addresses an important topic. One of the striking observations is the unequal impact of COVID-19 infection on men versus women and on older people versus younger people. The reasons for this remain unclear and any effort to shine a light on this would be beneficial to clinical practice and public health policy. The hypotheses explored in this review – about the role of sex hormones is interesting. Estrogen and testosterone are both known to mediate immune responses, with testosterone an immune suppressor and estrogen an immune booster, thus the speculation that the gender disparities may be related to hormonal differences are based on biological and physiological properties of the hormones. Whether hormonal status, due to age, treatment, or co-morbidities, influence COVID-19 morbidity is relevant.
I have a few suggestions that would greatly improve the report:
- Use of statistical evidence to support the hypothesis – although the authors cite a number of studies to support their hypothesis, they leave out or do not consistently report the statistical evidence. This could in the form of effect sizes, significance, or comparative data.
- On page 1, line 36 – in the previous line, they give the statistic for Italian patents who were male (82%), but when they report about the Canadian data, they report the % of women tested (63%), but not the % of males reported. This problem is repeated for the UK data. This information could be summarized in a table to make the case that men and women are being tested at the same rate, but the admission, severe disease, and mortality is higher in men. A graphical display could also help make this point.
- On page 1, line 41 – the authors report variable expression of ACE2, but do not provide the fold difference. Is this difference similar to the gender difference? How strong are these data?
- On page 2, lines 63-64 – it would be good to include the statistics, even if just as ranges.
- On page 2 lines 93 – it seems like the statistics cited have been inverted – please check. The authors should also note that the differences could due to differences in exposure to COVID-19 between the two groups.
- On page 3 line 110 – the authors speculate that genetic variants could explain the differences in COVID-19 fatality among countries – this is statement should be supported by some statistics of the genetic variants and a correlation of tat frequency with mortality. If such data do not exist, this statement should be included as pure speculation, or it should be deleted.
- On page 4, line 127 – has this effect been quantified?
- On page 7, line 178 – how strong are these correlations. These details are really important for assessing whether the differences are statistically and clinically meaningful.
- On page 2, line 45 – the inclusion of potential confounders is interesting. Is there any evidence that these factors influence ACE2 expression? If this is not known, the authors should say so. If known, please state the correlation.
- On page 2, line 49 – the authors introduce the issue of immune senescence and then in lines 53-54- the authors discuss the antibody levels b gender. These two issues should be introduced separately because senescence should work the same way in men and females, while the gender disparities should be greater in younger people.
- The idea that testosterone could be contributing in some form is intriguing – however, the mechanism is unclear. In general, testosterone is seen as an immunosuppressive agent by down-regulating natural killer cells and tumor necrosis factor-alpha. Is disease COVID-19 severity mediated by immunosuppression or by a strong pro-inflammatory response.
- Table 1 shows that there is interest in this area of research. However, this information could be given in a supplementary form. A more informative table would be a compilation of studies that have demonstrated effects – gender and age – the effect sizes, and level f significance, perhaps with some comments about the strength of the effects.
- On page 6 line 155 – the authors introduce the notion of “hypogonadism” – I think this notion could be introduced earlier when discussing age-related effects. However, the more interesting point here is that it is unclear whether “SARS-CoV-2 induces hypogonadism or not. In other words, is it known whether COVID-19 affects testosterone levels independent of age? Low testosterone levels could be due to aging hypogonadal males, which create a permissive environment lifts immunosuppressive role of testosterone?) for severe responses to COVID-19 infection or the virus inhibits androgen formation.